# Dynamic Neural Network is All You Need: Understanding the Robustness of Dynamic Mechanisms in Neural Networks

## Abstract

Deep Neural Networks (DNNs) have been used to solve different day-to-day problems. Recently, DNNs have been deployed in real-time systems, and lowering the energy consumption and response time has become the need of the hour. To address this scenario, researchers have proposed incorporating dynamic mechanism to static DNNs (SDNN) to create Dynamic Neural Networks (DyNNs) performing dynamic amount of computation based on the input complexity. Although incorporating dynamic mechanism into SDNNs would be preferable in real-time systems, it also becomes important to evaluate how the introduction of dynamic mechanism impacts the robustness of the models. However, there has not been a significant number of works focusing on the robustness trade-off between SDNNs and DyNNs. To address this issue, we propose to investigate four aspects of including dynamic mechanism into SDNNs. For that purpose, we evaluate four research questions. These evaluations are performed on three models and two datasets. Through the studies, we find that attack transferability from DyNNs to SDNNs is higher than attack transferability from SDNNs to DyNNs. Also, we find that DyNNs can be used to generate adversarial samples more efficiently than SDNNs. We also provide insight into the design choices through research studies. Finally, we propose a novel attack to understand the additional attack surface introduced by the dynamic mechanism.

## 1 Introduction

Deep Neural Networks (DNNs) are used in multiple applications such as computer vision and natural language processing. After the rapid growth of IoT and embedded devices, many real-time systems use DNNs in their applications. As the real-time systems require faster response time and low energy consumption, researchers have proposed to incorporate energy-saving dynamic mechanism (Wang et al., 2018; Kaya et al., 2019; Wu et al., 2018) to popular static DNN (SDNN) models like ResNet (He et al., 2015), VGG (Simonyan & Zisserman, 2014), MobileNet (Howard et al., 2017) etc. Early-exit is one of the dynamic mechanism techniques where multiple exits are included in SDNNs (creating multiple sub-networks), and SDNNs can terminate the operation early if a certain sub-network is confident about the prediction. These types of DNNs are named as early-exit Dynamic Neural Networks (DyNNs). Although the transition from SDNNs to DyNNs is preferred in real time systems because of increased efficiency, whether the use of dynamic mechanism will impact the robustness of the systems is still unknown. Studying the impact of the dynamic mechanisms on the robustness is important for developers or users to understand the trade-offs between DyNN and SDNN.

In this work, we propose to investigate four different aspects of including dynamic mechanism through four research questions. These four aspects are: *Transferability, Impact on Efficiency, Early-exits Design* and, *Added Attack Surface*.

**Transferability.** First, we investigate the adversarial attack transferability between SDNNs and DyNNs to evaluate the robustness of the models in black-box scenarios. In the black-box scenarios, adversaries normally assume the target models are always static. However, the target models can be

dynamic also. Hence, it is important to find out if a surrogate SDNN model is used to attack a target DyNN model or vice-versa, then, to which extent the adversary can be successful.

To address this issue, in this paper, we first conduct a comparative study on the adversarial attack transferability between SDNNs and DyNNs (Section 3). Our study results suggest that adversarial transferability from DyNNs to SDNNs is better and surprisingly using DyNNs as surrogate models for attack seems to be a more efficient and more effective way to generate adversarial samples. The adversaries are able to generate more adversarial samples in the same amount of time compared to using SDNNs as the surrogate model, and the generated adversarial samples often can also attack SDNNs.

**Impact on Efficiency.** Second, we conduct another study to understand whether the original purpose of DyNNs (*i.e.,* saving inference time) will be impacted by the adversarial samples (Section 4) generated through SDNNs. Our study results suggest that the adversarial samples generated by existing white-box attacks and black-box attacks do not increase the inference time significantly.

**Early-exits Design.** Third, we perform a detailed analysis of which design choices in the dynamic mechanisms or DyNN architectures (specifically position of early exits) may impact the robustness of DyNNs (Section 5). We consider two attack scenarios in this study: first, the output layer label of an SDNN is modified by a white-box adversarial example, and we study the impact of the example on corresponding DyNN's early-exit layers; second, in a black-box scenario, the output of SDNN is modified by a sample, and the sample is fed to separate model's DyNN. We have made multiple findings based on the empirical results, for example, putting the first exit earlier in the model architecture can help to improve the robustness of DyNNs.

**Added Attack Surface.** Last but not least, we design an adversarial attack approach to understand the extra attack surface introduced by the dynamic mechanisms in neural network (Section 6). In this attack, the synthesized adversarial examples will not change the prediction of the final output layer's label, but will change the prediction of all the early exits. Based on the attack results, we find that the dynamic mechanism is more vulnerable in scenarios where dependency among DyNN layers is lesser and when the exits are sparse w.r.t the layers.

## 2 RELATED WORKS AND BACKGROUND

**Dynamic Neural Networks.** The main objective of DyNNs is to decrease the energy consumption of the model inference for inputs that can be classified with fewer computational resources. DyNNs can be classified into *Conditional-skipping DyNNs* and *Early-exit DyNNs*. Early-exit DyNNs use multiple exits (sub-networks) within a single model and because of the model's working mechanism, the model is more suited for resource constrained devices. If, at any exit, the confidence score of the predicted label exceeds user defined threshold, inference is stopped. The resource-constrained devices usually deploy a lightweight sub-network of early exit network locally and resort to a server for further computations if needed (Teerapittayanon et al., 2017) . Graves (2016), Figurnov et al. (2017), Teerapittayanon et al. (2016), Kaya et al. (2019) have proposed Early-termination AdNNs. Specifically Kaya et al. (2019) and Zhou et al. (2020) propose early exit networks based on popular SDNNs. Zhou et al. (2020) also show that white-box robustness of the DyNNs is better than SDNNs. In addition to that multiple works (Teerapittayanon et al., 2017; Scardapane et al., 2020) provide practical usability of DyNNs.

Figure 1 shows the working mechanism of Early-exit DyNNs. For example, an Early-exit DyNN has N parts and each part has an exit. $x$ is the input, $f_{out}^i$ represents prediction after the $i^{th}$ part (generated by specific computation unit), $f_{out}$ represents prediction of the Neural Network, $C_i$ represents confidence score after $i^{th}$ part, $Hid_i^{In}$ represents input of $i^{th}$ part, $Hid_i^{Out}$ represents output of $i^{th}$ part, and $\tau_i$ is the predefined threshold to exit the network af-

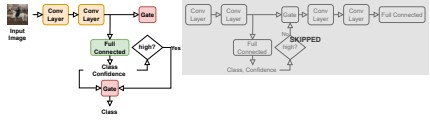

Figure 1: Working mechanism of Early-exit DyNN

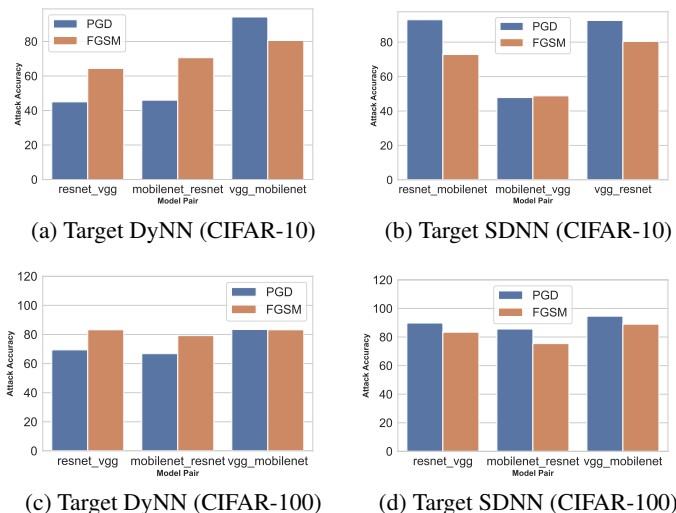

Figure 2: Transferable Attack Success Rate for CIFAR-10 and CIFAR-100 data

ter $i^{th}$ part. The working mechanism of the Early-exit DyNN can be represented as, $f_{out}(x) = f_{out}^i(x)$, if $C_i(x) \geq \tau_i$.

**Adversarial Attacks.** Adversarial Examples are the input that can change the prediction of the DNN model when those are fed to the model. Goodfellow et al. (2014) propose Fast Gradient Sign Method (FGSM) that uses single-step first order entropy loss to generate adversarial inputs. This attack is modified by Madry et al. (2017) to add initial noise to the benign sample. This attack is referred as projected gradient descent (PGD). Other than that, Dong et al. (2018); Carlini & Wagner (2017); Croce & Hein (2020); Lin et al. (2019) have proposed white-box attack methods, while Liu et al. (2016); Andriushchenko et al. (2020); Ilyas et al. (2018) have proposed black-box attack methods.

# 3 IS ADVERSARIAL EXAMPLE TRANSFERABILITY FROM DYNN TO SDNN IS LOWER THAN ADVERSARIAL EXAMPLE TRANSFERABILITY FROM SDNN TO DYNN?

In this research question, we investigate the "transferability" of adversarial inputs generated based on SDNN and DyNN, respectively, *i.e.,* whether adversarial examples generated based on SDNNs are adversarial to DyNNs and vice versa. Transferability is an important metric for evaluating the feasibility of black-box attack. To evaluate the transferability, one of the popular way (Papernot et al., 2017; Liu et al., 2016) is creating a similar model (*i.e.,* surrogate model) as the target model. In a black-box attack, normally, adversaries assume the underlying model to be SDNN, so for a deployed DyNN, the adversaries may likely use an SDNN as surrogate model. Hence, this research question (RQ) is important to evaluate the robustness of DyNNs.

## 3.1 EXPERIMENTAL SETUP.

**Dataset and Models.** We use CIFAR-10 and CIFAR-100 (Krizhevsky et al., 2009) datasets for evaluation. For SDNNs, we use VGG-16 (Simonyan & Zisserman, 2014), ResNet56 (He et al., 2015), and MobileNet (Howard et al., 2017) model. As DyNNs, we use the early exit version of these models (Kaya et al., 2019). In all other RQs, we keep the models and dataset setup same.

**Black-box Attack.** For the attack scenario, we use surrogate model (Papernot et al., 2017; Liu et al., 2016) based black-box attack scenario. Here, we feed a set of inputs to the target model and collect the output labels. These inputs are generated using 50% of the held-out validation data, and naturally corrupted versions of those validation data. As the number of partial held-out validation

data is not significant, adding corrupted inputs would help to increase training data size for the surrogate model. For natural corruption (Hendrycks & Dietterich, 2019), we use gaussian noise and brightness. For each type of corruption, we have five intensity levels. For example, if the number of held out data is 5000, for each corruption, we generate 25000 additional data. Once the input-output pairs are collected, a surrogate model is trained based on those pairs. For a target model, we use both SDNNs and DyNNs. If the target model is SDNN, then an DyNN is trained as surrogate model and vice-versa.

To make the surrogate-target pairs, we use different types of DNN architectures. For example, if the the target model is DyNN VGG, we choose ResNet56 SDNN as the surrogate model. This assumption is valid because the attacker doesn't have information about the target model architecture, hence the possibility of choosing the same architecture as the surrogate model is less. We define two terms to represent two different types of transferability based on different types of surrogate model and target model: **D2S** transferability and **S2D** transferability. D2S transferability evaluates DyNN to SDNN attack transferability, where S2D transferability evaluates SDNN to DyNN attack transferability. We have chosen following pairs to evaluate S2D transferability: (SDNN ResNet56 (surrogate), DyNN VGG (target)), (SDNN MobileNet (surrogate), DyNN MobileNet (target)), (SDNN MobileNet (surrogate),DyNN ResNet56 (target)). Similarly, for D2S transferability, earlier mentioned surrogate models become the target model and earlier mentioned target models become the surrogate model.

**Algorithms.** We use FGSM (Goodfellow et al., 2014) and PGD (Madry et al., 2017) algorithms to attack the surrogate models.

**Metric.** We measure percentage of adversarial examples that can mis-classify the output w.r.t number of generated adversarial examples as the attack success rate.

### 3.2 EVALUATION RESULTS

Figure 2 shows the effectiveness of black-box attacks on DyNNs and SDNNs. On average, it can be noticed that for target SDNN and surrogate DyNN, the attack success rate is higher than the success rate of target AdNN and surrogate SDNN. One of the reasons for this behavior is the lower variance of the DyNNs. DyNNs use lower number of parameters, hence the feature space for adversarial samples of DyNNs is smaller than the feature space for adversarial samples of SDNNs (Schönherr et al., 2018). Also, we find that for the target DyNN, FGSM attack performs better than PGD attack. If only dataset-specific results are considered, then for CIFAR-100 the attack success rate is higher than for CIFAR-10.

Also, as the DyNN-generated adversarial inputs can attack SDNNs, then it can be time efficient to create adversarial inputs using DyNN. Through Figure 7 (Appendix), we can find the probability density plots of different exit numbers of DyNN that have been used to generate adversarial examples. Lower exit number suggests that lesser number of computations has been used to generate adversarial examples. It can be noticed that for PGD attack, more than 70% of adversarial examples are generated from exit 0, 1 and 2 (first - third exit). Although for FGSM attack, in a few scenarios (CIFAR-10 VGG, CIFAR-100 MobileNet, and CIFAR-100 ResNet), more than 50% of adversarial examples are generated through later exits (higher computation required).

> **Finding 1**: *The **D2S** transferability is higher than the **S2D** transferability.*
> **Finding 2**: *Using DyNNs as surrogate models is more efficient and more effective way to generate adversarial examples than using SDNNs.*

## 4 DOES ADVERSARIAL EXAMPLES IMPACT EARLY-EXIT EFFICIENCY IN DYNNS?

In this section, we investigate whether the original purpose of including dynamic mechanism in DyNNs (*i.e.,* saving inference time) will be impacted by the adversarial samples. Specifically, we study whether the adversarial inputs exit earlier or later in a DyNN compared to the original inputs. The main objective of this investigation is to find out whether the adversarial samples generated on

SDNN can have an impact on the amount of computation of the DyNN. For this purpose, we conduct both white-box and black-box attacks to find out the impact of adversarial samples on the amount of computation.

Here, the white-box attack scenario can also be considered a practical scenario. There have been studies (Chen et al., 2022; Wu et al., 2022) that focus on reverse engineering of SDNN models from binary code of on-device models, but no techniques have been proposed to reverse engineer the dynamic mechanisms in the models. Hence, adversaries are more likely to get SDNN models instead of their dynamic counterparts. So it is important to find out how adversarial samples affect the efficiency of the DyNN for both white-box and black-box scenarios.

### 4.1 EXPERIMENTAL SETUP.

**Attack.** We use PGD and FGSM for both white-box and black-box attacks. For black-box setup, we use the same setup as previous RQ. For black-box scenario, we use DyNNs as target model and SDNNs as surrogate model. In a white-box setting, we attack the SDNN and evaluate on the performance of corresponding DyNN.

**Metric.** We use the difference between the exit number selected by adversarial input and the exit number selected by benign input. If the difference is positive, then the latency of the adversarial sample is increased w.r.t benign input.

### 4.2 EVALUATION RESULTS.

Figure 3 and Figure 10 (in Appendix) show the impact of adversarial examples generated on SDNN on changing in exit number in DyNN in a white-box setting. It can be observed that for the majority of the scenarios, accuracy-based adversarial samples do not increase the computation significantly in the DyNN. On average, FGSM-generated examples increase more computation in DyNNs than PGD-generated examples. For CIFAR-10 data, for MobileNet and VGG-16 DyNN models, 25%-37% of the FGSM generated examples could increase the number of exits by more than one. Also, it can be noted that adversarial samples generated on CIFAR-100 data is more likely to increase computation than adversarial samples generated on CIFAR-10. Especially, more than 45% of the FGSM samples generated on CIFAR-100 data can increase the number of exits by more than one.

Figure 4 and Figure 11 (in Appendix) show the impact of adversarial examples generated on SDNN on change in exits in DyNN in a black-box setting. It can be noted that, for CIFAR-10 dataset, black-box attack can generate more computation-increasing examples than in white-box attack. For ResNet and VGG model, more than 40% PGD attack generated examples can increase the number of exits by more than one. For all three models, 35% of the FGSM attack generated examples can increase the number of exits by more than one. FGSM attack generates more inputs that induces low confidence in early-exit layers than PGD attack. For CIFAR-100 dataset, the increase of computation caused by adversarial attacks is higher than that of CIFAR-10. As CIFAR-100 data uses larger model, the robustness of the model is reduced. For CIFAR-100 dataset, more than 40% examples generated through both the attack can increase the number of exits by more than one. However, increasing the number of exits by two or three exits does not decrease the efficiency in DyNNs significantly.

> **Finding 3**: *Accuracy-based adversarial samples do not decrease the efficiency significantly in the DyNN.*
> **Finding 4**: *The adversarial examples, whose output confidences are significantly lower, can perform better in terms of decreasing the efficiency in DyNNs.*
> **Finding 5**: *Adversarial examples generated on a larger model (w.r.t model parameters) is more likely to decrease efficiency in DyNNs.*

## 5 WHAT DESIGN OF DYNNS MAY IMPACT THE ROBUSTNESS?

In this section, we evaluate which architecture design choices (position of early exits) in DyNNs may impact the robustness of early layers. In this RQ, we consider DyNNs as multi-exit networks, where

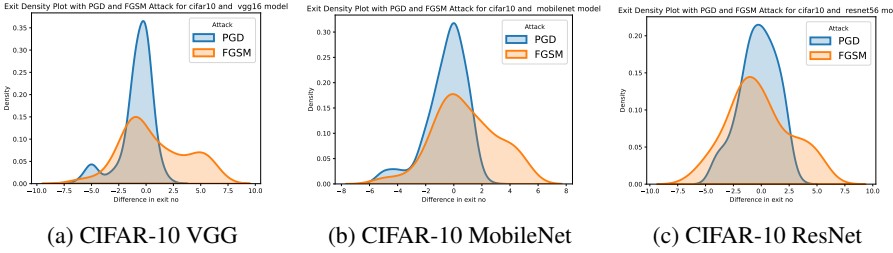

(a) CIFAR-10 VGG  (b) CIFAR-10 MobileNet  (c) CIFAR-10 ResNet

Figure 3: Density plots of change in exit numbers because of PGD and FGSM attack (For CIFAR-10 data). The x axis represents the change in exit number while y axis represents the density.

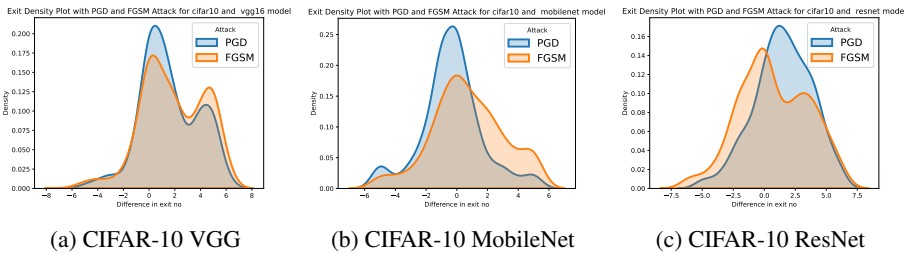

(a) CIFAR-10 VGG  (b) CIFAR-10 MobileNet  (c) CIFAR-10 ResNet

Figure 4: Density plots of change in exit numbers because of PGD and FGSM **black-box** attack (For CIFAR-10). The x axis represents the change in exit number while y axis represents the density.

each exit will provide an output. For evaluation, we assess if we attack the final exit, from which early-exit layer the label modification begin. If the output label is not modified in any earlier layer, then for that type of model, the robustness is higher because the model can produce correct results at any layer. This RQ will provide us an insight into which type of design choice may improve the robustness of early layers.

### 5.1 EXPERIMENTAL SETUP.

**Attack Setup.** We use same attack setup as previous research questions. However, while we attack the DyNN, we do not consider one specific exit layer. Instead of that, we consider each exit layer.

**Metric.** We use the early exit from which the label modification starts. For example, there are $N$ exits. First, $N$th exit's label is modified through adversarial sample. If till $K$th exit the original prediction was same, then we report $K + 1$th exit in the experimentation.

### 5.2 EVALUATION RESULTS.

Figure 5 and Figure 8 (in Appendix) show probability density plot on which exit the output label is changed using the white-box adversarial examples. It can be observed that for all the model-dataset pairs, for more than 77% of the examples, the label is modified in the first exit. For CIFAR-10 data, only for MobileNet and ResNet models, the label is changed after the first exit for more than 20% of the examples (using FGSM attack).

Figure 6 and Figure 9 (in Appendix) show probability density plot on which exit the output label is changed using the black-box adversarial examples. From results, we can see that the robustness of earlier exits is better against black-box attack than white-box attack. For CIFAR-10 data, more than 45% of the both attack generated samples could not misclassify the first exit for VGG-16 model. For CIFAR-100 data (larger model), robustness of early exit layers is worse than that of CIFAR-10 data (smaller model). For CIFAR-100 data, for both attacks and three models, less than 30% of the adversarial examples can not mis-classify the first exit.

For black-box attack scenario, VGG-16 model's first layer on-average shows better robustness against black-box attack. In the DyNN, the first exit of the VGG model is placed only after sec-

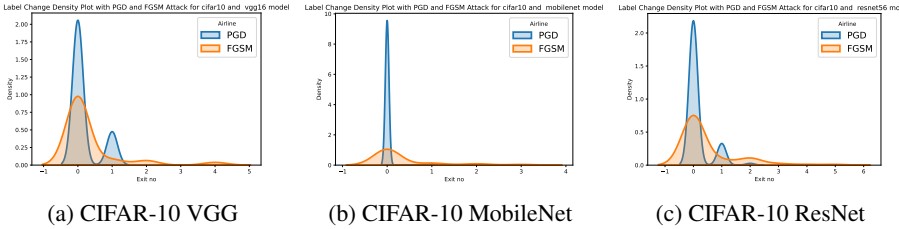

Figure 5: Density plots representing during which exit number output label is changed because of PGD and FGSM attack (For CIFAR-10 data). The x axis represents the exit number while y axis represents the density.

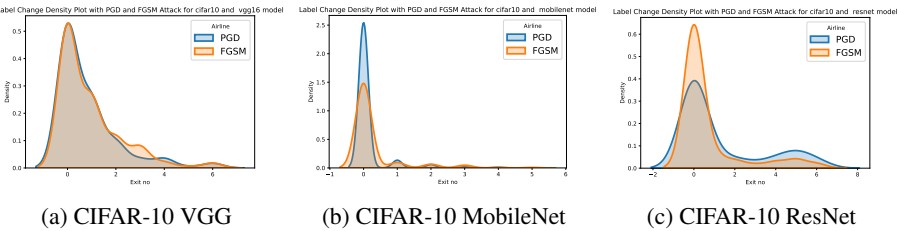

Figure 6: Density plots representing during which exit number output label is changed because of PGD and FGSM **black-box** attack (For CIFAR-10 data). The x axis represents the exit number while y axis represents the density.

ond layer, while for others, more computations are performed before using the first exit. Hence, having the first exit in the early layers can increase the robustness of DyNN. Although lower number of parameters would be used to predict output, but with VGG we can notice that a significant number of inputs can be classified correctly through first exit.

> **Finding 6**: *We find that having the first exit in the earlier layers can increase the robustness of early exits of the DyNNs.*
> **Finding 7**: *Black-box attack success rate is lesser than white-box attack success rate against early-exit layers.*
> **Finding 8**: *For black-box scenario, early-exit layers are more robust against adversarial examples generated on a smaller surrogate model than adversarial examples generated on a larger surrogate model.*

# 6    CAN WE SYNTHESIZE CERTAIN EXAMPLES TO EVALUATE THE ADDED ATTACK SURFACE OF DYNAMIC MECHANISM?

In this section, we evaluate if specific examples can be generated only to understand the additional attack surface introduced by the dynamic mechanism in DyNNs. We aim to design an attack such that the synthesized adversarial examples will not change the prediction of the final SDNN label, but change the prediction of all the early exits. This threat model is practical because the attack evades the existing detection that relies on the final output of SDNN while the attacker creates a situation where all the early exit networks do not provide correct prediction, therefore decreasing the usability of DyNNs. However, this attack is also challenging to be performed successfully because final layer output is dependent on the earlier exit layers and it is challenging to impact all the early exits' prediction without modifying the final exit's prediction.

## 6.1    PROBLEM FORMULATION

We propose a novel attack technique called *Early Attack* to evaluate the effectiveness of the early layers of DyNNs. Let's assume $f$ is an DyNN with $N$ exits. Given an input $x$, the output softmax layer in an exit $i$ can be defined as $y^i = f_i(x)$, where $i = 1, 2, 3...N$. For synthesizing adversarial examples, we have two main objectives. First, the initial prediction in the $N^{th}$ exit (final layer) does not change. Let's assume, the initial prediction at final layer is $p$. Second, the prediction of all the other exits should be different than $p$.

---

**Algorithm 1:** Input generation using Early Attack

---

**Inputs** : $x : Input\ Image$
**Outputs:** $x' : Perturbed\ Image$

1 **begin**
2     Initialize($w$)
3     $T = number\_of\_iterations$
4     $iter\_no = 0$
5     **while** $iter\_no < T$ **do**
6         $x' = \frac{\tanh(w)+1}{2}$
7         $output = model(x')$
8         **if** $success(output)$ **then**
9             $return(x')$
10         **end**
11         $L = loss(x, w, c, \alpha)$
12         $L_{new}, w = Optim(L, w)$
13         $iter\_no + +$
14     **end**
15     $x' = \frac{\tanh(w)+1}{2}$
16 **end**

---

Based on the aforementioned objectives, we can propose an iterative optimization procedure to optimise a loss function $L$. For each of the objectives, one loss function component is formulized. For the first objective, we propose loss function $L1 = (-1 * \sum_{j \neq p} max(y_p^N - y_j^N, 0))$. In $L1$, we maximize the difference between softmax value of label $p$ and other label's softmax value, therefore the prediction won't get changed at the final exit. For the second objective, we propose loss function $L2 = (\sum_{i=1}^{N} y_p^i)$. The $L2$ ensures that for any other exit than the final exit, $p$'s softmax value would be minimized. The final loss function $L = \alpha * L1 + L2$. Here the $\alpha$ is a user-defined variable that provides balance between two loss terms.

Finally, we need to ensure the added perturbation to generate adversarial examples are limited, hence, the final optimization function would be, $minimize(||\delta|| + c \cdot L)$, where, $(x + \delta) \in [0, 1]^n$. Here, $\delta$ represents the added perturbation and $c$ is a user-defined variable to provide weightage on a specific component. $c$ controls the magnitude of generated perturbation ($||\delta_i||$), where a large $c$ makes the loss function more dependant on the $L$.

This constrained optimization problem in $\delta$ can be converted into a non-constrained optimization problem in $w$, where the relationship between $\delta$ and $w$ is: $\delta = \frac{\tanh(w)+1}{2} - x$ The $tanh$ function would ensure that the generated adversarial input values stay between 0 and 1. The equivalent optimization problem in $w$ is:

$$\underset{w}{minimize} \left\| \frac{\tanh(w) + 1}{2} - x \right\| + c \cdot L \tag{1}$$

Algorithm 1 shows the optimization algorithm. The algorithm outputs the adversarial input $x'$ given a benign image $x$ as input. $w$ is initialized to a random tensor that has equal shape as the input image. For each iteration, the loss function of the attack is computed (at line 11). Based on the back-propagated loss, the optimizer updates $w$ with its next value. Once the iteration threshold ($T$) is reached or the attack is successful, the algorithm computes and returns the adversarial input $x'$ (at Line 9 and 15).

## 6.2 EVALUATION

### 6.2.1 EVALUATION SETUP.

**Baseline.** We use PGD and FGSM attacks as baseline to modify the early exit label.

**Metric.** We use attack success rate as metric in the evaluation. If the adversarial input generated final layer output is same as the output generated by benign input and all the other exit layers output is different than the final output label, then we consider attack is successful for that particular adversarial input.

**Hyperparameters.** We use $\alpha = \{0.001, 0.01, 0.1, 1, 20, 40\}$ and $c = 50$ as hyperparameters.

### 6.2.2 EFFECTIVENESS

Table 1 shows the evaluation results of the attack success rate of Early Attack and baseline techniques. Except two model-dataset pair, Early Attack's success rate is higher than 80% in all other

scenarios. PGD and FGSM attacks are unsuccessful because the loss function only considers early layer output and does not consider final layer output. Although FGSM's attack success rate is higher than PGD's success rate. Also, w.r.t best performing $\alpha$ values, $\alpha = 1$ performs well for 70% of the model-dataset pairs.

We also analyze why early attack fails for VGG16 and MobileNet models for CIFAR-10 data. First, for CIFAR-10 data, final fully connected layer has lower number of parameters than the final layer for CIFAR-100 data. Hence, the dependency between the final layer and earlier layers is higher for CIFAR-10. However, for CIFAR-10 data, attack success rate is higher for ResNet model. First we discuss the failed cases for VGG16 and MobileNet model, and then we discuss why the attack succeeded for ResNet model.

For VGG16 model, the failed examples can be divided into two types. For the first type, the final output label is changed with all the other early exit layers. In the second type, first few layers get the output label mis-classified, but along with the final layer, few previous layers also get the output correctly classified. For MobileNet, all the layers except the final and last to final layer get the output label mis-classified. For MobileNet, we also find that the final two exits are separated by only two layers, which is significantly lower than other model's separation layers between final two exits. Hence, the dependency between final two layers is higher.

For ResNet model, there are 56 layers divided into 27 blocks. The exits are sparsely divided between these blocks. For Mobilenet and VGG models, the exit distributions are less sparse. Hence, the dependency between exits is lower for ResNet and because of that reason, we could successfully attack ResNet model.

### 6.2.3 TRANSFERABILITY

In this section, we discuss about the transferability of the Early Attack examples. As Early Attack has two different components, instead of measuring attack success rate directly, we measure two parameters $T1$ and $T2$. $T1$ represents the percentage of inputs for which the final output remains same as the output generated by benign input. $T2$ represents from the examples selected from $T1$, how many early exit layers on average are misclassified. Having both high $T1$ and $T2$ values ensures transferability.

Table 1: **Attack accuracy percentage of Early Attack and the baseline techniques against different model and dataset, along with $\alpha$ value.**

| Dataset | Model | Early Attack | best $\alpha$ val | PGD | FGSM |
|---------|-------|--------------|-------------------|-----|------|
| CIFAR-10 | VGG | **35** | 1 | 0 | 0 |
| | MobileNet | **11** | 0.1 | 0 | 0 |
| | ResNet | **81** | 1 | 0 | 0 |
| CIFAR-100 | VGG | **86** | 20 | 0 | 1 |
| | MobileNet | **97** | 1 | 0 | 0 |
| | ResNet | **96** | 1 | 0 | 2 |

We show the transferability results in Appendix (Table 2). For CIFAR-10 data, the $T1$ values are high, but $T2$ values are significantly low except for MobileNet to ResNet transferability. For CIFAR-100 data, $T2$ values are higher than of CIFAR-10 data, however, $T1$ values are low. From the results, we can notice that the generated examples either can keep the final layer label the same or can change the early exit layer outputs. Our finding suggests that early attack transferability is limited.

> **Finding 9**: *With increasing number of exits in DyNNs, the dependency between multiple exits will increase. Hence, more exits in DyNNs can increase the robustness against the Early Attack.*
> **Finding 10**: *Early Attack transferability between DyNNs is not significant.*

## 7 CONCLUSION

In this work [1], we discuss the robustness of including dynamic mechanism in DNN through four research questions. We find out that DyNNs are more robust than SDNNs and also efficient to generate adversarial examples. We also propose DyNN design choices through final two RQs. Finally, we propose a novel attack to understand additional attack space in DyNNs.

---

[1]https://github.com/anonymous2015258/Early_Attack/tree/main

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

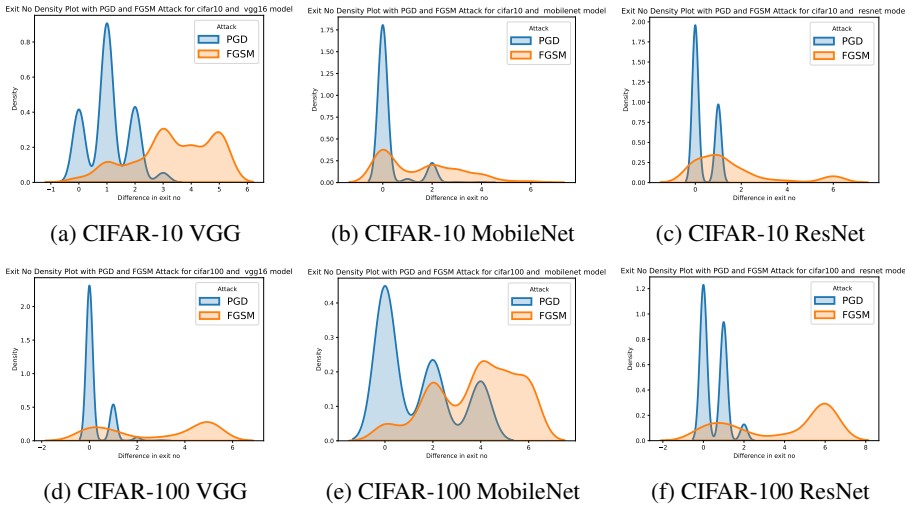

Figure 7: Density plots of exit numbers in DyNN that are used to generate black-box adversarial inputs using PGD and FGSM attacks. The x axis represents the exit number while y axis represents the density.

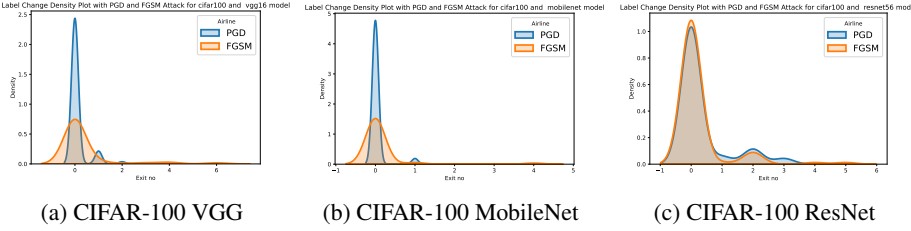

Figure 8: Density plots representing during which exit number output label is changed because of PGD and FGSM attack (For CIFAR-100 data). The x axis represents the exit number while y axis represents the density.

# Appendices

## A  RQ1 DENSITY PLOTS

The density plots in Figure 7 shows the probability density plots of exit numbers in DyNN that are used to generate black-box adversarial inputs using PGD and FGSM attacks.

## B  EARLY ATTACK SUCCESS RATE WITH DIFFERENT $\alpha$ VALUES

Table 2: **Attack accuracy percentage of Early Attack with different $\alpha$ values**

| Dataset | Model | EA($\alpha$=0.001) | EA($\alpha$=0.01) | EA($\alpha$=0.1) | EA($\alpha$=1) | EA($\alpha$=20) | EA($\alpha$=40) |
|---------|-------|--------|--------|--------|--------|--------|--------|
| CIFAR-10 | VGG | 0 | 0 | 0 | 35 | 10 | 4 |
| | MobileNet | 0 | 0 | 11 | 4 | 0 | 0 |
| | ResNet | 7 | 32 | 73 | 81 | 49 | 32 |
| CIFAR-100 | VGG | 1 | 5 | 48 | 82 | 86 | 70 |
| | MobileNet | 0 | 16 | 74 | 97 | 92 | 77 |
| | ResNet | 46 | 74 | 92 | 96 | 96 | 93 |

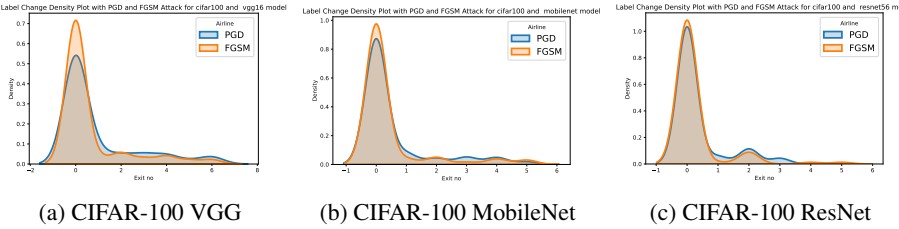

(a) CIFAR-100 VGG  (b) CIFAR-100 MobileNet  (c) CIFAR-100 ResNet

Figure 9: Density plots representing during which exit number output label is changed because of PGD and FGSM **black-box** attack (For CIFAR-100 data). The x axis represents the exit number while y axis represents the density.

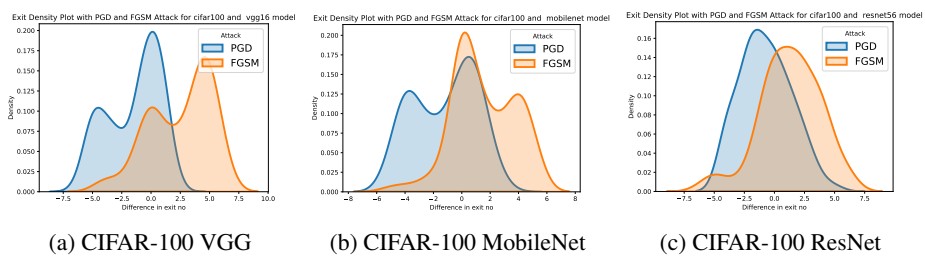

(a) CIFAR-100 VGG  (b) CIFAR-100 MobileNet  (c) CIFAR-100 ResNet

Figure 10: Density plots of change in exit numbers because of PGD and FGSM attack (For CIFAR-100 data). The x axis represents the change in exit number while y axis represents the density.

## C  TRANSFERABILITY

Table 2 shows the $T1$ and $T2$ values of all three models on two datasets.

## D  RQ1 RESULTS BASED ON MI-FGSM ATTACK

Here, through Figure 12, we show the adversarial transferability results between DyNNs and SDNNs using MI-FGSM attack. These results again confirm that adversarial examples from DyNN to SDNN are more transferable than adversarial examples from SDNN to DyNN.

## E  TRANSFERABILITY EXPERIMENTS ON MI-FGSM ATTACK

Through Figure 12, we show the S2D and D2S transferability with MI-FGSM attack Dong et al. (2018). The results confirm our claim that D2S transferability is higher than S2D transferability.

Table 3: $T1$ and $T2$ values for measuring transferability between three models. *TSM* represents Target Model and *SM* represents Surrogate Model. $T1$ presents the percentage of inputs for which the final output remains same as the output generated by benign input. $T2$ represents from the examples selected from $T1$, how many early exit layers on average is mis-classified.

| Type | SM ＼ TM | CIFAR-10 | | | CIFAR-100 | | |
|---|---|---|---|---|---|---|---|
| | | VGG | MNet | RNet | VGG | MNet | RNet |
| $T1$ | VGG | – | 85% | 73% | – | 65% | 39% |
| | MNet | 86% | – | 74% | 51% | – | 38% |
| | RNet | 89% | 82% | – | 79% | 75% | – |
| $T2$ | VGG | – | 0.73 | 1.65 | – | 1.58 | 2.69 |
| | MNet | 1.06 | – | 2.3 | 1.52 | – | 3.26 |
| | RNet | 0.95 | 0.85 | – | 1.32 | 1.28 | – |

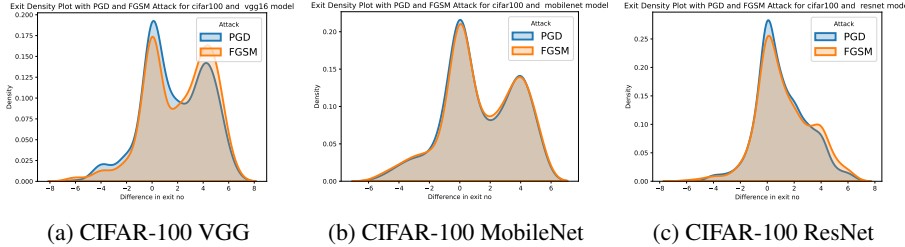

(a) CIFAR-100 VGG      (b) CIFAR-100 MobileNet      (c) CIFAR-100 ResNet

Figure 11: Density plots of change in exit numbers because of PGD and FGSM **black-box** attack (For CIFAR-100 data). The x axis represents the change in exit number while y axis represents the density.

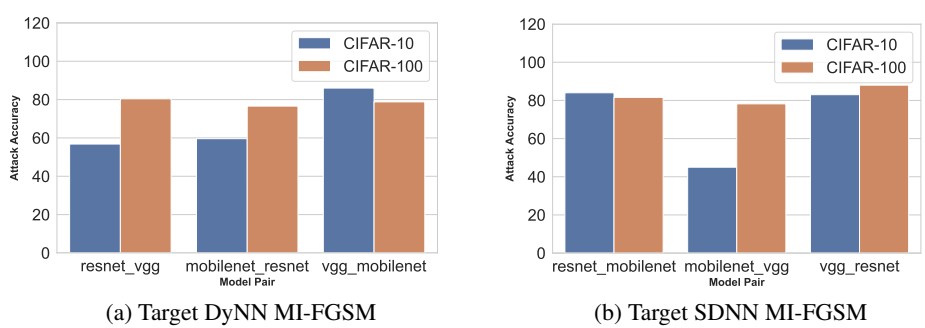

(a) Target DyNN MI-FGSM      (b) Target SDNN MI-FGSM

Figure 12: Transferable Attack Success Rate for MI-FGSM attack

## F    COMPARING DIFFERENT ADVERSARIAL IMAGES

In this section, we show original images, adversarial images generated through DyNNs and adversarial images generated through SDNNs through Figure 13, Figure 15 and Figure 14. We find that in terms of quality, images generated through SDNNs (Average PSNR (Fardo et al., 2016) = 23.20) are slightly better than images generated through DyNNs (Average PSNR = 23.19).

## G    RQ1 RESULTS BASED ON TINY IMAGENET IMAGES

Here, through Figure 16, we show the adversarial transferability results between DyNNs and SDNNs for Tiny Imagenet (Chrabaszcz et al., 2017) datasets. These images are larger in size ($64 \times 64$) than CIFAR images (size $32 \times 32$). These results reconfirm that adversarial examples from DyNN to SDNN are more transferable than adversarial examples from SDNN to DyNN, even if the input feature space is larger. Although, we can notice a slight decrease in transferability for both D2S and S2D.

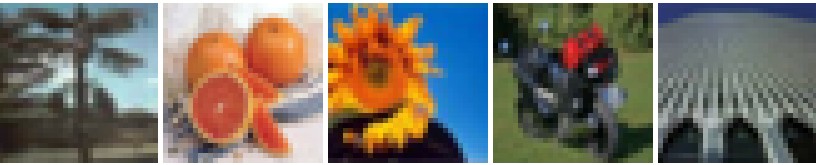

Figure 13: Original Images

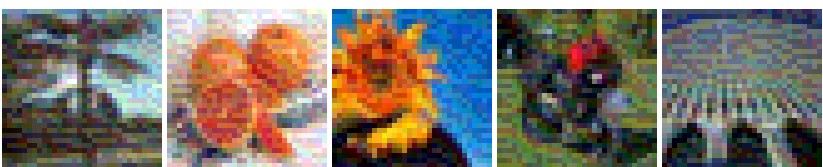

Figure 14: Adversarial Images generated on SDNNs

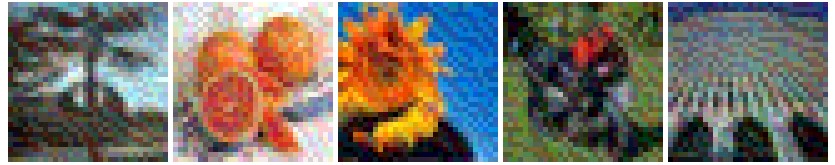

Figure 15: Adversarial Images generated on DyNNs

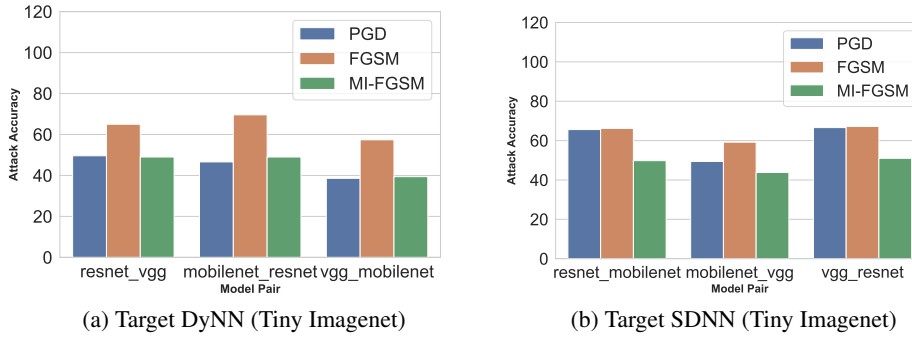

(a) Target DyNN (Tiny Imagenet)      (b) Target SDNN (Tiny Imagenet)

Figure 16: Transferable Attack Success Rate for Tiny Imagenet Data

