# OpenReview forum: "Dynamic Neural Network is All You Need: Understanding the Robustness of Dynamic Mechanisms in Neural Networks"
_ICLR.cc/2023/Conference — Submitted to ICLR 2023_

### Official Review · Reviewer_X4b3 · 2022-10-24

**Confidence:** 4
**Correctness:** 3
**Technical Novelty And Significance:** 4
**Empirical Novelty And Significance:** 4
**Recommendation:** 8

**Clarity, Quality, Novelty And Reproducibility:**

Overall the paper is easy to read and clearly sets the research questions that it addresses.

The following is not properly justified. Does this follow an established understanding by the community?
“One of the reasons for this behavior is the lower variance of the DyNNs. DyNNs use lower number of parameters, hence the feature space for adversarial samples of DyNNs is smaller than the feature space for adversarial samples of SDNNs.”

It would be interesting to show the adversarial examples generated by DyNNs vs static networks. Are there particular differences that could be visually interesting?

The Appendix contains some useful information that should be in the main manuscript. For instance, Figure 7 and Algorithm 1. I suggest the authors carefully review the structure of the paper in order to try and include the additional content within the page limit.


**Strength And Weaknesses:**

Strengths:
- I like the questions raised in this paper and believe they provide a new direction for research in dynamic neural networks.
- The experiments use some standard deep learning models like VGG and ResNet, and explore different scenarios e.g., either dynamic or static model as surrogate etc.
- Some interesting findings such as that the more exits increase the robustness of the model.

Weaknesses:
- The related works section needs to be expanded significantly in order to provide the reader with a more complete picture especially with the available works on dynamic neural networks.
- The list of references is also quite limited considering that adversarial attacks are a very popular topic and dynamic neural networks have been investigated for some years.
- The evaluation datasets include only cifar-10 and 100. It is not clear of using other benchmark such as ImageNet which have higher resolution would impact the results since now a much larger attack surface is available.
- It would have been nice to see a theoretical framework specific for dynamic neural network that would provide some additional foundations besides the empirical findings.


**Summary Of The Paper:**

This paper investigates the robustness of dynamic neural networks, focusing specifically on early exit networks. It does so by exploring how attacks generated using static networks transfer to dynamic networks and how the structure of dynamic networks provides another attack surface. In their majority most works appearing in the literature dynamic networks have been utilized to improve real-time performance. However, they have not been investigated under the spotlight of robustness. In this regard I find that this work provides some interesting findings which have not been explored previously.

**Summary Of The Review:**

The paper investigates the robustness of dynamic neural networks against adversarial attacks. It provides a new focus for dynamic neural networks and highlights the new attack surface that they provide. While this paper has some shortcomings and can be improved in some areas I would recommend it based on the new aspects that it tries to highlight.

---

> ### Author Response · Authors · 2022-11-17
> **Addressing the Concerns of Reviewer X4b3**
>
> We thank the reviewer for the comments. Here is our explanation.
>
> **Justifying Claim in RQ1:**   Although we do not have any theoretical proof about the claim in RQ1, our claim is based on the fact that ‘simpler models are more robust [1].’ As the number of parameters used by DyNN is lesser than the parameters used by SDNN, the variance of the DyNN is lower, making it more robust. Previous adversarial attacks like Schönherr et al [2] have also mentioned that with more parameters, adversaries get more adversarial space to explore.
>
> **Additional Changes:** We will make the following changes in the draft based on the suggestions.
>
> We have added adversarial images based on DyNN and SDNN in the appendix (Figure 13 to 15). We also calculated the Peak Signal to Noise Ratio of these images, which shows that both adversarial images are similar w.r.t quality.
> We have added more works in  the reference section.
> We are adding a few experiments on an additional dataset. As the time is limited, we are considering using tiny-imagenet. We will add that in the draft.
> Restructuring the paper to include Figure 7 and Algo 1 to main draft.
>
> [1] Skiena, Steven S. The data science design manual. Springer, 2017.
>
> [2] Schönherr, Lea, et al. "Adversarial attacks against automatic speech recognition systems via psychoacoustic hiding." arXiv preprint arXiv:1808.05665 (2018).

---

> > ### Author Response · Authors · 2022-11-18
> > **Evaluation on larger input surface**
> >
> > **Evaluation on Tiny Imagenet.** We evaluate the transferability between SDNNs and DyNNs (RQ1) on the Tiny Imagenet dataset. The images on this dataset are larger in size (64 $\times$ 64) than CIFAR images (size 32 $\times$ 32).
> > The results shown in Figure 16 reconfirm that adversarial examples from DyNN to SDNN are more transferable than adversarial examples from SDNN to DyNN, even if the input feature space is larger.

---

### Official Review · Reviewer_bhB9 · 2022-10-24

**Confidence:** 3
**Clarity, Quality, Novelty And Reproducibility:** None
**Correctness:** 2
**Technical Novelty And Significance:** 2
**Empirical Novelty And Significance:** 2
**Recommendation:** 3

**Strength And Weaknesses:**

### Strength:

1. The paper is well-written with summarized findings in each section (3-6).
2. Experiments are conducted in different architectures and datasets.
3. Sufficient implementation details are provided.

### Weakness:

1. Why do the authors only consider early-exit DyNNs? "more popular" seems not convincing.
2. The logical relationship between these four perspectives of DyNNs' robustness is not clear, which results in this paper being more like a technical report rather than a conference paper.
3. The literature on adversarial attacks missing a lot of important works. e.g., Auto-Attack, C&W attack, etc.
4. Finding 1 in Section 3 "With respect to black-box attacks, early-exit DyNNs are more robust than SDNNs" is really debatable. 1) missing white-box robustness evaluation 2) Only FGSM and PGD attack is not reliable for assessing robustness. With more advanced attacks, e.g., Auto-Attack, does this finding also hold?


**Summary Of The Paper:**

The work studies the robustness of dynamic neural networks (DyNNs) from four perspectives: 1) comparison of the black-box robustness between static and DyNNs; 2) The inference time of adversarial samples in DyNNs; 3) Dynamic mechanisms with the best robustness; 4) synthesize samples to reduce the effectiveness of DyNNs.

**Summary Of The Review:**

None

---

> ### Author Response · Authors · 2022-11-17
> **Addressing the Concerns of Reviewer bhB9**
>
> We thank the reviewer for the comments. Here is our explanation.
>
> **Weakness 1.** The main usage of DyNNs is through the application of Early-exit DyNNs on resource-constrained devices[1,2,8,9]. The resource-constrained devices usually deploy a lightweight sub-network locally and resort to server computation as needed. Early-exit DyNNs are preferable in this setting because all inputs coming into the DyNNs need to execute the first sub-network. So developers can deploy just the first sub-network of a large DyNN locally on the device and put the other parts of the model on the server. The other type of DyNNs (i.e., skipping DyNNs) requires deploying the whole large model on the device.
>
> [1]Scardapane, Simone, et al. "Why should we add early exits to neural networks?." Cognitive Computation 12.5 (2020): 954-966.
>
> [2] Phuong, Mary, and Christoph H. Lampert. "Distillation-based training for multi-exit architectures." Proceedings of the IEEE/CVF International Conference on Computer Vision. 2019.
>
> [8] Kaya, Yigitcan, Sanghyun Hong, and Tudor Dumitras. "Shallow-deep networks: Understanding and mitigating network overthinking." International conference on machine learning. PMLR, 2019.
>
> [9] Teerapittayanon, Surat, Bradley McDanel, and Hsiang-Tsung Kung. "Branchynet: Fast inference via early exiting from deep neural networks." 2016 23rd International Conference on Pattern Recognition (ICPR). IEEE, 2016.
>
>
>
>
> **Weakness 2.**  We focus on studying *the trade-off in introducing dynamic mechanism to a DNN* in this work. For that study, we have designed four RQs that specifically evaluate four components of this trade-off.
>
> *Transferability.*  First, in RQ1, we focus on comparing the transferability in DNNs **with** the dynamic mechanism and **without** the dynamic mechanism and find that transferability in DNNs with the dynamic mechanism is lower than transferability in DNNs without the dynamic mechanism.
>
> *Impact on efficiency.* Second, in RQ2, we focus on how **the efficiency of dynamic mechanisms** is impacted by adversarial examples.
>
> *Early-exits design.* Then, in RQ3, we evaluate **different design choices** for dynamic mechanisms (specifically the position of early exits) to evaluate robustness in early layers against adversarial examples.
>
> *Added attack surface.* And last, in RQ4, we discuss **new** attack surfaces exposed by introducing dynamic mechanisms to specifically show trade-off between using and not using dynamic mechanisms.
>
>
>
> **Weakness 3.** We have added references in the draft.
>
> **Weakness 4.**  We have modified our claim in RQ1 to: DyNN-to-SDNN attack **transferability** is higher than SDNN-to-DYNN attack transferability.
> In addition, our paper actually investigated the white-box attack scenario in RQ4, where we evaluate the attack space added through the inclusion of dynamic mechanisms in DyNNs. The key difference here is that we focus on added attack surface of  introducing dynamic mechanisms to DNNs instead of just the attack surface of a model.
>
>
>
> *Additional Experimentation.* For additional experimentation (as suggested by the reviewer), we have used a newer adversarial attack named MI-FGSM [7], which can generate adversarial examples with higher transferability than popular whitebox attacks [4,5,6]. The experimental results are provided in the appendix (Figure 12), which confirms our claim that adversarial examples from DyNN to SDNN are more transferable than adversarial examples from SDNN to DyNN.
>
>
>
> [3]Huang, Yi, and Adams Wai-Kin Kong. "Transferable Adversarial Attack based on Integrated Gradients." arXiv preprint arXiv:2205.13152 (2022).
>
> [4] Ian J Goodfellow, Jonathon Shlens, and Christian Szegedy. “Explaining and harnessing adversarial examples. International Conference on Learning Representations”, 2015.
>
> [5] Alexey Kurakin, Ian Goodfellow, and Samy Bengio. “Adversarial examples in the physical world.” International Conference on Learning Representations, Workshop Track Proceedings, 2017
>
> [6] Nicholas Carlini and David Wagner. “Towards evaluating the robustness of neural networks.” In 2017 ieee symposium on security and privacy (sp), pages 39–57. IEEE, 2017.
>
> [7]Dong, Yinpeng, et al. "Boosting adversarial attacks with momentum." Proceedings of the IEEE conference on computer vision and pattern recognition. 2018.

---

> > ### Comment · Reviewer_bhB9 · 2022-11-26
> > **Thanks for the detailed rebuttal**
> >
> > Many thanks to the authors for the detailed rebuttal. Some of my problems have been well addressed and I've raised my score accordingly.
> >
> > But I still have one concern about the reliability of the evaluated robustness. It would be great if the authors could assess the robustness through Auto-Attack (https://github.com/fra31/auto-attack).

---

> > > ### Author Response · Authors · 2022-12-01
> > > **Evaluation on Auto-attack**
> > >
> > > | Att                       | RN (SDNN) |  RN (DyNN) | MN (SDNN) |  MN (DyNN) | VGG (SDNN) |  VGG (DyNN) |
> > > | -------------         | -------          | ------------- | --------------| -------------- | ---------------|  ---------------|
> > > | APGD-DLR         |92%                    |90%                    |94%                      |94%                     |94%                      |93%                       |
> > > | APGD-CE           |92%                    |90%                    |95%                      |90%                      |92%                      |90%                      |
> > > | Fab                     |95%                    |90%                    |92%                      |90%                      |95%                      |93%                      |
> > > | Square         	     |44%                    |3%                    |29%                      |7%                      |16%                      |8%                      |
> > >
> > > We thank the reviewer for raising the score. Based on the suggestion, we perform the robustness evaluation of DyNN and SDNN models on Auto-attack [1].  Auto-attack consists of four different attack techniques: apgd-dlr, apgd-ce, fab and square. While the first three attack techniques are whitebox, the square attack technique is performed in a black-box scenario. We evaluate the robustness on the CIFAR-10 dataset. The attack success rates (ASR) of the attacks are shown in the aforementioned table.
> > >
> > > **Evaluation on Black-box Scenario.** Square attack is a query-based attack that is implemented  without accessing the model. For square attack, the ASR values are significantly higher in SDNN models than DyNN models. The evaluation solidifies the claim that in blackbox scenario, DyNN models are more robust than SDNNs.
> > >
> > >  **Evaluation on White-box Scenario.** For all three white-box attacks, both SDNN and DyNN models show high ASR. However, ASRs against DyNN models are slightly lower than ASRs against SDNN models.
> > >
> > > [1] Croce, Francesco, and Matthias Hein. "Reliable evaluation of adversarial robustness with an ensemble of diverse parameter-free attacks." International conference on machine learning. PMLR, 2020.

---

> > > > ### Author Response · Authors · 2022-12-03
> > > > **Auto-attack Experimentation Result on CIFAR-100 dataset**
> > > >
> > > > | Att                       | RN (SDNN) |  RN (DyNN) | MN (SDNN) |  MN (DyNN) | VGG (SDNN) |  VGG (DyNN) |
> > > > | -------------         | -------          | ------------- | --------------| -------------- | ---------------|  ---------------|
> > > > | APGD-DLR         |74%                    |69%                    |74%                      |68%                     |68%                      |68%                       |
> > > > | APGD-CE           |74%                    |69%                    |74%                      |68%                      |68%                      |68%                      |
> > > > | Fab                     |74%                    |69%                    |74%                      |72%                      |70%                      |69%                      |
> > > > | Square         	     |50%                    |10%                    |20%                      |12%                      |28%                      |10%                      |
> > > >
> > > >
> > > > We also performed the auto-attack experiments on the CIFAR100 dataset. We notice that for white-box attacks, the ASR values are lower across different models than ASR values evaluated for CIFAR-10 data. Although for both white-box and black-box attacks, ASR values against DyNNs are lower than SDNNs. Hence, the results indicate that the robustness of DyNNs is higher than the robustness of SDNNs against auto-attack.

---

### Official Review · Reviewer_nmxr · 2022-10-24

**Confidence:** 4
**Correctness:** 4
**Technical Novelty And Significance:** 2
**Empirical Novelty And Significance:** 3
**Recommendation:** 6

**Clarity, Quality, Novelty And Reproducibility:**

Clarity: Good
Quality: Good
Novelty: OK
Reproducibility: Good

**Strength And Weaknesses:**

Strengthes:
1. The adversarial robustness of DyNNs is an important research problem. Some of the findings are new and can be interesting to the community.
2. The paper is overall well-written and easy to follow.
3. The experiments are well conducted and well supports the findings in the paper.

Weaknesses:
1. Missing related work on DyNN with improved adversarial robustness [Zhou et al. 2020]. This work also investigated the adversarial robustness of DyNNs and introduced a novel early-exit criteria for improved robustness. This makes the novelty of some findings less strong. However, [Zhou et al. 2020] does not conduct extensive experiments and the throughout empirical analysis in this paper can still be beneficial.
2. The proposed early-attack method is not very well motivated. It's unclear in which scenario one would expect the prediction of last layer to be unchanged because this will not influence the final prediction anyway.

[Zhou et al. 2020] BERT Loses Patience: Fast and Robust Inference with Early Exit


**Summary Of The Paper:**

This paper presents an empirical study of the adversarial robustness of dynamic neural networks (DyNNs) with early-exits. The authors find that DyNNs are more robust than SDNNs, and DyNNs can be used to generate adversarial samples efficiently. The authors also proposes a novel adversarial attack method specifically designed for DyNNs.

**Summary Of The Review:**

This paper conducts an interesting empirical study about the adversarial robustness of DyNNs. It misses an important related work and is thus somewhat overclaiming the novelty of the findings. Nevertheless, some findings are still interesting and can be helpful.

---

> ### Author Response · Authors · 2022-11-10
> **Addressing the Concerns of Reviewer nmxr**
>
> We thank the reviewer for the comments. Here is our explanation.
>
> **Comparison with Bert Loses Patience:** We thank the reviewer for pointing out the related work. While we focus on studying *the trade-off between DNN with the dynamic mechanism and without the dynamic mechanism*, Bert Loses Patience proposes a new DyNN to increase the robustness of the DyNN model. Although we understand that our first finding might have similarity with Bert Loses Patience’s conclusion, our work provides multiple general findings regarding the trade-off between DNN with and without the dynamic mechanism.
>
> First, in RQ1, we focus on comparing the *transferability in DNNs with the dynamic mechanism and without the dynamic mechanism* and find that transferability in DNNs with the dynamic mechanism is lower than transferability in DNNs without the dynamic mechanism, and it is more efficient to use dynamic mechanism to generate adversarial examples for DNNs without the dynamic mechanism.
>
> Second, in RQ2, we focus on how *the efficiency of dynamic mechanisms is impacted by adversarial examples*. Our finding states that examples generated through larger models are more likely to decrease efficiency of dynamic mechanisms.
>
> Then, in RQ3, we evaluate the design choices for dynamic mechanisms to *increase robustness in early layers against adversarial examples*. And last, in RQ4, we discuss new attack surfaces exposed by introducing dynamic mechanisms to specifically show trade-off between using and not using dynamic mechanisms. Also, we provide design choices to resist this type of attack.
>
>
> **Usability of Early Attack:** Through early attack, we evaluate new attack surfaces exposed by using dynamic mechanisms in existing SDNNs. Our evaluation shows for which model architectures, attacking early layers would or would not have an impact on the final layer. *The reason we want to exclude adversarial inputs that can already attack the final layer is that we want to exclude the old attack surface already existing in the SDNNs.* Finally, we provide design choices based on the evaluation.

---

> > ### Comment · Reviewer_nmxr · 2022-11-30
> > **Response to authors' response**
> >
> > Thanks for the response. Some of my concerns are partially resolved and I will keep my score of borderline acceptance.

---

### Author Response · Authors · 2022-11-17
**Current and Future Modifications to the draft**

Reviewers, here are the changes we have made to the draft.

1. We have improved the related works section.

2. We have moved the early attack algorithm and early-exit dynn overview image to the main draft from the appendix.

3. We have added DyNN and SDNN generated adversarial inputs to the appendix (Fig 13-15).

4. We have added transferability experiments on a more transferable attack (MI-FGSM). (Fig 12)

Here are the additional changes to the draft we are making and will resubmit soon.

1. We are modifying the draft's introduction and abstract to explain our objective in a better way.

 2. We are adding RQ1 experiments with tiny-imagenet, where the image size is larger than CIFAR-10 and CIFAR-100.

---

> ### Author Response · Authors · 2022-11-18
> **Modified Version of Draft**
>
> Reviewers, we have uploaded a new version of the draft will following changes:
>
> 1. We have modified the draft's introduction and abstract to explain our objective in a better way.
> 2. We have added RQ1 experiments with Tiny Imagenet, where the image size is larger than CIFAR images.

---

### Author Response · Authors · 2022-12-06
**Gentle Ping**

Dear Reviewers, As the stage 2 discussion is going to end soon, please let us know if any more doubts/queries we need to address.

---

### Decision · Program_Chairs · 2023-01-20

**Decision:**

Reject

**Justification For Why Not Higher Score:**

See justification for why not lower score.



**Justification For Why Not Lower Score:**

This paper is "yet another adversarial example paper in a slightly different setting." Other than the usual academic deference given to the potential practical consequences of (1) adversarial examples and (2) dynamic neural networks, I don't think the work is well-motivated. Why do we care about adversarial examples in the context of dynamic neural networks? Is this a science paper that helps us gain some deeper understanding of adversarial robustness by looking at the problem a different way? Is there something distinctly different in practice about dynamic neural networks that motivates a different view of adversarial examples? This paper is a technical report where everything appears to be true, but I don't see a clear motivation for why it matters. It's a strong technical report, so I'm ambivalent about whether it is accepted.

**Metareview: Summary, Strengths And Weaknesses:**

**Summary:** This paper looks at adversarial robustness in the context of early-exit neural networks. The paper entails a number of different adversarial methods, transfer attacks, etc. - the standard battery of things one does when dealing with adversarial robustness.

**Strengths:** The paper is an excellent catalogue of experiments on dynamic neural networks. The experiments are comprehensive and well chosen.

**Weaknesses:** At the end of the day, this is a technical report cataloging existing properties on a new (and pretty niche) artifact. It's not particularly novel or significant. It's more of a "findings" paper than something that will inspire new research directions.

**Recommendation:** I have recommended a very ambivalent acceptance. I would not be disappointed if the paper were rejected on the basis of lack of significance. I leave it to the senior AC to decide whether this meets the threshold.

**Summary Of Ac-Reviewer Meeting:**

N/A